Genome-wide identification and functional analysis of class III peroxidases in Gossypium hirsutum

Chen Yi
Feng Jiajia
Qu Yunfang
Zhang Jinlong
Zhang Li
Liang Dong
Yang Yujie
Huang Jinling huangjlsxau@163.com
College of Agriculture, Shanxi Agricultural University , Taigu, Shanxi , China
Adhikari Tika
Electronic publication date: 2022 Jul 1
Publication date: 2022
Volume: 10
Electronic Location ID: e13635
Received 2021 Dec 30; Accepted 2022 Jun 6
Copyright: © 2022 Chen et al.
Copyright year: 2022
Copyright holder: Chen et al.
License: This is an open access article distributed under the terms of the Creative Commons Attribution License, which permits unrestricted use, distribution, reproduction and adaptation in any medium and for any purpose provided that it is properly attributed. For attribution, the original author(s), title, publication source (PeerJ) and either DOI or URL of the article must be cited.
License URL: https://creativecommons.org/licenses/by/4.0/

Keywords: Gossypium hirsutum, Class III peroxidase, Pollen fertility, VIGS, Overexpression

Funding: Special Project of Academic Restoration and Scientific Research of Shanxi Agricultural University 2020xshf40 National key R & D programs 2018YFD0100301 Doctoral Research Project of Shanxi Agricultural University 2020BQ46 This work was funded by the Special project of academic restoration and scientific research of Shanxi Agricultural University (2020xshf40), the National Key R & D programs (2018YFD0100301), and the Doctoral research project of Shanxi Agricultural University (2020BQ46). The funders had no role in study design, data collection and analysis, decision to publish, or preparation of the manuscript.

==============================
Class III peroxidase (PRX) genes play essential roles in various processes, such as auxin catabolism, removal of H2O2, crosslinking cell wall components, and response to biotic and abiotic stresses. In this study, we identified 166, 78 and 89 PRX genes from G. hirsutum, G. arboretum and G. raimondii, respectively. These PRX genes were classified into seven subfamilies based on phylogenetic tree analysis and the classification of PRX genes in Arabidopsis. Segmental duplication and purifying selection were the major factors driving the evolution of GhPRXs. GO and KEGG enrichment analysis revealed that GhPRX genes were mainly associated with responding to oxidative stresses, peroxidase activities and phenylpropanoid biosynthesis pathways. Transcriptome data analysis showed that GhPRX genes expression were significantly different in microspore development between the sterility line-JinA and the maintainer line MB177. We confirmed the up-regulation of GhPRX107 and down-regulation of GhPRX128 in the sterile line compared to its maintainer line using qRT-PCR, suggesting their roles in pollen fertility. In addition, silencing GhPRX107 in cotton showed a significant decrease of the reactive oxygen species (ROS) levels of microsporocyte stage anthers compared to control. Overexpressing GhPRX107 in Arabidopsis significantly increased the ROS levels of anthers compared to wild type. In conclusion, we identified GhPRX107 as a determinant of ROS levels in anther. This work sets a foundation for PRX studies in pollen development.

Introduction

Peroxidase (EC 1.11.1.X) family is an important enzyme family which can catalyze oxidoreduction (Dunford & Stillman, 1976). These enzymes have been widely found in animals, plants, and microorganisms. Peroxidases are classified into three major types based on protein structures and catalytic properties: class I (ascorbate peroxidase), class II (lignin peroxidase) and class III (secretory peroxidase) (Bhatt & Tripathi, 2011).

Class III peroxidases (PRXs, EC 1.11.1.7) are plant-specific. They are encoded by multiple gene, 73 in Arabidopsis (Tognolli et al., 2002), 93 in Populus and 138 in Rice (Passardi et al., 2004). Class III peroxidase family members contain disulfide bridges, calcium ions and n-terminal signal peptides (Sundaramoorthy et al., 1994). PRXs are glycosylated and located in extracellular spaces or vacuoles (Jacobowitz, Doyle & Weng, 2019).

PRXs are not only thought to function by oxidizing target substrates with hydrogen peroxide (H2O2), but also act as key factors in producing reactive oxygen species (ROS). There are two main kinds of biochemical pathways catalyzed by PRXs in plant; the first pathway is the oxidation pathway. PRXs catalyzes the reduction of hydrogen peroxide (H2O2) by using different substrates such as lignin precursors, phenolic compounds, and secondary metabolites as the electron donor in this pathway (Hiraga et al., 2001; Passardi, Penel & Dunand, 2004). The second pathway is the carbonylation cycle. PRXs can catalyze the production of ROS by participating in the secondary metabolic carbonylation cycle (Liszkay, Kenk & Schopfer, 2003).

PRXs are important for plant growth and development. They are involved in many important biological activities and have multiple isozymes with distinct catalytic properties (Passardi et al., 2005). PRXs participate in a broad range of physiological processes such as auxin catabolism (Hiraga et al., 2001), removal of H2O2 (Ostergaard et al., 2000), lignin biosynthesis (Lee et al., 2013), crosslinking of cell wall components, and stress responses (Marjamaa et al., 2006) by oxidizing target molecules (Sasaki et al., 2004) and regulating ROS levels (Chen & Schopfer, 1999). PRXs respond to biotic stresses. For example, OsPrx30 encodes a secretory protein located in multiple organelles. Overexpression of OsPrx30 enhanced plant susceptibility to rice bacterial blight by maintaining high levels of POD activity and reducing H2O2. It showed the opposite effect when the expression of OsPrx30 was suppressed (Liu et al., 2021). When cotton plants were infected with Verticillium dahlialis (Dong et al., 2019), PRX genes responded by changing their expressions. PRXs are also involved in plant growth and development. Ghpox1 participated in cotton fiber elongation and development by mediating ROS production (Mei et al., 2009). In addition, PRXs are involved in male reproductive processes of plant. PRX9 and PRX40 have been identified to be essential for the normal development of tapetum and microspore in Arabidopsis. The PRX9/PRX40 double mutant showed unique tapetum swelling and pollen grain enlargement, which resulted in microspore degeneration and male sterility (Jacobowitz, Doyle & Weng, 2019). Ghpod gene is specifically expressed in floral organs from a single recessive male sterile line of G. hirsutum, indicating that PRX may be related to male fertility development of cotton (Chen et al., 2009). Interestingly, PRXs can exert opposite biological functions: some produce ROS while others scavenge ROS; some loosen the cell wall while others reinforce the cell wall (Shigeto & Tsutsumi, 2016).

So far, expression patterns and functions of PRXs have rarely been reported due to the complexity of the PRX gene family. In order to explore the role of PRXs in male reproductive processes of cotton, we studied PRXs in G. hirsutum, G. arboretum and G. raimondii and determined their phylogenetic relationships. We used bioinformatics methods to analyze chromosome locations, cis-acting elements, and expression patterns of G. hirsutum PRX genes. We predicted the functions of PRX genes in G. hirsutum using GO and KEGG enrichment analysis. We validated GhPRX107 function by virus-induced gene silencing (VIGS) in G. hirsutum and overexpression in Arabidopsis. Our results set a foundation for further studying the role of PRXs in male reproductive processes of cotton.

Materials and Methods

Plant materials

The cotton cytoplasmic male sterility line (JinA) and its maintainer line (MB177) were used to explore the patterns of temporal expression in flower buds. The sterility and maintainer line plants were planted in Shanxi Agricultural University. Flower buds of different stages (sporogonium stage, microsporocyte stage and meiosis stage) were collected and stored at −80 °C for subsequent RNA extraction and qRT-PCR analysis. We used maintainer line (MB177) for gene silencing. The maintainer line plants were planted in artificial climate chamber (23 °C, 70% relative humidity, photoperiod of 8 h darkness/16 h light) and cotton flower buds were collected at Microsporocyte stage for subsequent qRT-PCR and anthers ROS level analysis. The wild-type Arabidopsis thaliana Columbia (Col-0) was used for gene overexpression. They were planted in artificial climate chamber (22 °C, 60% relative humidity, photoperiod of 8 h darkness/16 h light). The samples were collected at different development stage flower buds (6, 7, 8, 9, 10, 11, 12 stages) for subsequent anthers ROS level analysis.

Identification of PRX family members in Gossypium hirsutum, G. arboretum and G. raimondii

The genome sequences and annotation files of G. hirsutum (Wang et al., 2019) (TM-1 HAU_v1.1), G. arboretum (Wang et al., 2021) (HAU_v1.0), and G. raimondii (Paterson et al., 2012) (JGI, v2.0) were downloaded from CottonGen (https://www.cottongen.org/) (Yu et al., 2014). The protein sequences of Arabidopsis were obtained from the Uniprot (https://www.uniprot.org) (Pundir, Martin & O’Donovan, 2017). Taking the Arabidopsis PRXs (Tognolli et al., 2002) as reference sequences, the whole genome protein sequences of three cotton species were scanned using the BLASTP program (e-value <1e−5) of TBtools (Chen et al., 2020). All identified PRXs were confirmed the existence of the conserved domains using NCBI CDD (http://www.ncbi.nlm.nih.gov/cdd) (Lu et al., 2020). The redundant sequences were removed using CD-Hit (Fu et al., 2012) with default parameters. The isoelectric point (pI) and molecular weight (MW) of PRXs were calculated using ExPASy (https://www.expasy.org/) (Duvaud et al., 2021). The signal peptide of PRXs were predicted using SignalP (http://www.cbs.dtu.dk/services/SignalP/) (Nielsen, 2017). The subcellular localizations of PRXs were predicted using WoLF PSORT (https://wolfpsort.hgc.jp/) (Horton et al., 2007).

Multiple alignments and phylogenetic analysis

The full amino acid sequences of PRXs from three cotton species and Arabidopsis were aligned using clustalW program (Thompson, Higgins & Gibson, 1994). The parameters for alignment by clustalW were as follows: gap opening penalty, 10; gap extension penalty, 0.2; protein weight matrix, gonet; residue-specific penalties, on; hydrophilic penalties, on; delay divergent cutoff (%): 30. A maximum likelihood (ML) phylogenetic tree was constructed using MEGA 7.0 program (Kumar, Stecher & Tamura, 2016) with bootstrap 1,000 repetitions and the Jones-Taylor-Thornton (JTT) model (Jones, Taylor & Thornton, 1992), then was drawn using EvolView (He et al., 2016). The PRXs in G. hirsutum were assigned to different subfamilies based on phylogenetic relationships and the classification of PRXs in the Arabidopsis (Tognolli et al., 2002).

Chromosome locations and gene structural of PRX genes in Gossypium hirsutum

The genome and the General Feature Format 3 (GFF3) files of G. hirsutum (Wang et al., 2019) (TM-1 HAU_v1.1) were downloaded from CottonGen (https://www.cottongen.org/) (Yu et al., 2014). The GFF3 file contains annotation information for the location of genes, coding sequences (CDS), and untranslated regions (UTRs) in the genome. According to the genome and annotation files of G. hirsutum, we obtained and visualized the gene structure (exons/introns) information and chromosomal positions of PRXs using TBtools (Chen et al., 2020).

Gene duplication and calculation of Ka/Ks values

To identify the duplication events that occurred in GhPRXs of the G. hirsutum genome, the whole genome sequences of G. hirsutum were compared using the BLASTP program (e-value <1e−10) of TBtools (Chen et al., 2020). Then, the MCScanx (Wang et al., 2012) with default parameters was used to detect the duplication patterns including segmental and tandem duplication. TBtools (Chen et al., 2020) was used to visualized paralogous gene pairs. In order to understand the selection pressures during the expansion of GhPRX gene family, the nonsynonymous mutation rate (Ka), synonymous mutation rate (Ks), and Ka/Ks values of homologous gene pairs were calculated by KaKs Calculator program in TBtools (Chen et al., 2020).

Analysis of cis-acting element in promoters

Upstream region of 1,500 bp from the translation initiation codon ATG of GhPRXs were selected as the promoter and entered into the Plantcare website for promoter analysis (http://bioinformatics.psb.ugent.be/webtools/plantcare/html/) (Lescot et al., 2002).

Analysis of GO and KEGG pathway enrichment

For functional enrichment analysis, the Gene Ontology (GO) and Encyclopedia of Genes and Genomes (KEGG) pathway enrichment analysis were performed using the Omicshare tools (https://www.omicshare.com/tools), taking false discovery rate (FDR) ≤ 0.05 as a threshold. The genes in the background file used by GO and KEGG were all genes that have been annotated to the GO term and KEGG pathway, from cotton genetic improvement group of HuaZhong Agricultural University (Wang et al., 2019) (Table S1).

Expression pattern analysis

To further explore functions of GhPRXs in G. hirsutum, the RNA-seq data of eight different tissues (bract, petal, torus, root, leaf, stem, pistil, sepal and anther) of G. hirsutum (TM-1) were downloaded from the NCBI (https://www.ncbi.nlm.nih.gov/) (Sayers et al., 2021) (accession number: PRJNA490626). The transcriptome data of cytoplasmic male sterility line-Jin A and maintainer line MB177 flower buds were obtained from Shanxi Agricultural University cotton breeding laboratory (Yang, Han & Huang, 2014b). Trimmomatic (Bolger, Lohse & Usadel, 2014) was used to perform quality control and remove the adapters. Specifying parameters were as follows: adaptors were considered based on sequencing instrument as default, “SLIDINGWINDOW is 4:15 and minimum read length is 30 bp”. Those remaining were aligned to the G. hirsutum (Wang et al., 2019) (TM-1 HAU_v1.1) genomes using the hisat2 program (Kim, Langmead & Salzberg, 2015), then Cufflinks (Ghosh & Chan, 2016) was used to calculate the fragments Per Kilobase of transcript per Million fragments (FPKM) values. The calculating parameters of Cufflinks were as follows: frag-bias-correct and multi-read-correct were used in this step. GhPRXs with FPKM > 1 were considered as expressed genes. TBtools software (Chen et al., 2020) was used to visualize the expression patterns of the GhPRXs based on the value of log2 (FPKM + 1).

RNA extraction and qRT-PCR analysis

We used EASYspin plant RNA quick isolation kit (RN38; Aidlab Biotech, Beijing, China) to extract the whole RNA. The Takara Rverse Transcription kit (Japan) was used to generate the first cDNA strand. The specific primers were designed using the NCBI database (https://www.ncbi.nlm.nih.gov/) (Sayers et al., 2021) (Table S2). We used the CFX96 Real-Time PCR Detection System (Bio-Rad, Hercules, CA, USA) to test qRT-PCR. The final volume of 20 μL including 10 μL SYBR Green PCR mix (Takara, Maebashi, Japan), 1 μL of specific primers, 1 μL of cDNA and 7 μL of ddH2O. The reaction program was as follow: one cycle of 95 °C for 30 s; 40 cycles of 94 °C for 5 s; one cycle of 60 °C for 30 s, and 40 cycles of 72 °C for 30 s. The housekeeping EF-1α gene was used as the reference (Yang, Han & Huang, 2014b). The experiment was repeated three times. We calculated expression levels of GhPRX genes using 2−∆∆Ct method (Rao, Lai & Huang, 2013). The statistical test was performed using t-tests (Livak & Schmittgen, 2001), and P < 0.05 was considered indicating a statistically significant difference (*P < 0.05; **P < 0.01).

VIGS

In order to explore the function of GhPRX107 in G. hirsutum, we used the tobacco rattle virus (TRV)-based vectors to preform virus-induced gene silencing (VIGS) (Pang et al., 2013). The specific primer was designed using SnapGene software (from Insightful Science; available at snapgene.com) (Table S3). We cloned the highly specific region of 400-bp from GhPRX107 into the EcoRI and BamHI sites of the TRV-based (pYL156) vector using ClonExpress® II One Step Cloning Kit (Vazyme Biotech Co, Ltd, Nanjing, China) to generate the TRV:GhPRX107 vector. The plasmid of TRV2:GhPRX107 and TRV2:00 vectors were transformed into Agrobacterium tumefaciens GV3101 and subsequently transformed into the maintainer lines-MB177 by cotyledon injection. The photobleaching phenotype which silenced the GhCLA1 gene in MB177 by VIGS was used as phenotype control. We used the qRT-PCR to detect the silencing efficiency of GhPRX107 gene in TRV2:GhPRX107 plant, the TRV2:00 lines were used as control. We analyzed anthers ROS (O2− and H2O2) levels of microsporocyte stage in control (TRV2:00) and TRV2:GhPRX107 plants by staining with nitroblue terazolium (NBT) and 3,3-diaminobenzidine (DAB), respectively.

GhPRX107 overexpression in Arabidopsis

To further explore the function of GhPRX107, we manipulated GhPRX107 levels by overexpression of GhPRX107 in Arabidopsis. We cloned GhPRX107 coding sequence from anthers cDNA using the primer pair PRI-F/PRI-R (Table S3). It was integrated into the PRI-AN-101 vector at the EcoRI and XbaI sites using ClonExpress® II One Step Cloning Kit (Vazyme Biotech Co, Ltd, Nanjing, China) to generate the PRI-GhPRX107 vector. The plasmid of PRI-GhPRX107 was transformed into Agrobacterium tumefaciens GV3101 and subsequently transformed into wild type Arabidopsis by floar dip method (Pang et al., 2013). We compared anthers ROS (O2− and H2O2) levels of different development stages (Sanders et al., 1999) between wild type and overexpression A.thaliana by staining with nitroblue terazolium (NBT) and 3,3-d iaminobenzidine (DAB), respectively.

Results

Identification of PRX family members in Gossypium hirsutum, G. arboretum and G. raimondii

In order to identify the PRX family members in G. hirsutum, G. arboretum and G. raimondii, we used the 73 Arabidopsis PRX proteins (Tognolli et al., 2002) as a reference to search and blast proteins from the three cotton genomes. In addition, we used NCBI CDD (http://www.ncbi.nlm.nih.gov/cdd) (Lu et al., 2020) to confirm if every PRX member contained the complete conserved domain of PRXs. After eliminating redundant sequences, we identified a total of 166, 78 and 89 PRX genes in G. hirsutum, G. arboretum and G. raimondii, respectively (Table S4). The correspondinrg PRX genes were renamed based on the chromosomal locations.

The predicted isoelectric point (pI) and molecular weight of PRX proteins were 4.07–10.43 (MW, 33.25–126.85 kDa) in G. arboretum, 4.13–10.46 (MW, 21.09–40.58 kDa) in G. raimondii, and 4.07–10.84 (26.67–70.61 kDa) and 4.13–10.88 (25.64–49.67 kDa) at the At and Dt subgenomes in G. hirsutum (Table S4), suggesting physical property differences between diploid cotton (G. arboretum and G. raimondii) and tetraploid cotton species (G. hirsutum). We observed that most PRX proteins were predicted to contain a signal peptide (294/333, 88.28%), which aligned with the properties of secreted proteins. The subcellular localization prediction results showed that 48.94% of PRXs located in chloroplasts, 27.92% in extracellular spaces, and the remaining 23.11% in cytoplasm, nucleus, vacuoles, and mitochondria (Table S4). The diverse set of predicted organelle locations implied different functions of PRX members.

Phylogenetic analysis

To understand the evolutionary relationships of the PRX gene family, we constructed a maximum-likelihood (ML) phylogenetic tree by repeating PRX proteins from G. hirsutum, G. arboretum, G. raimondii and Arabidopsis for 1,000 times. Combining the results of phylogenetic tree analysis with those of previous studies in Arabidopsis, we categorized PRXs into seven subfamilies (Fig. 1). Each subfamily contained PRX genes of the four species, indicating this gene family was conserved in different species during evolution. In addition, we observed that the homology of the PRX sequences was high among most of the PRXs derived from the At subgenome of the allotetraploid cotton (G. hirsutum) and the PRXs from G. arboretum. The PRX sequences from the Dt subgenome of G. hirsutum had high homology with the PRX genes from G. raimondii. This was consistent with the hypothesis that the allotetraploid cotton species came from the recombination of two diploid cotton species (Liu et al., 2015). The AtPRX9, AtPRX40 and GhPRX89 belonged to the same subfamily. Previous studies showed that GhPRX89 was involved in the male reproductive processes of cotton (Chen et al., 2009). AtPRX9 and AtPRX40 have been identified to be essential for the Arabidopsis anther development (Jacobowitz, Doyle & Weng, 2019). Collectively the results suggested that genes in the same subfamily share similar functions.

Figure 1 Phylogenetic analysis of PRX proteins from G.hirsutum, G. arboretum, G. raimondii and Arabidopsis.

The PRX proteins from G. hirsutum, G. arboretum, G. raimondii and Arabidopsis were marked with check, circle, triangle and star, respectively.

Chromosome locations of PRX genes in Gossypium hirsutum

Based on the annotated and sequencing information of G. hirsutum, we constructed a chromosomal map (Fig. 2), where 162 GhPRX genes were unevenly distributed on the 26 G. hirsutum chromosomes. Four PRX genes were found on the scaffold (Fig. 2). Among the chromosome-located genes, 79 and 83 were on the At- and Dt-subgenome chromosomes, respectively. The number of PRXs in allotetraploid cotton (G. hirsutum) was not equal to the diploid species (G. arboretum and G. raimondii), which is likely due to either gene loss in tetraploid species or gain in diploid species after polyploidization event. For the At subgenome, most genes were located on A05 (n = 11), and A13 had the least number of genes (n = 1); For the Dt subgenome, most gene were located on D12 (n = 12), and D06 had the least number of genes (n = 2). Additionally, we observed that some chromosome regions exhibited a relatively higher density of GhPRX genes, such as the bottoms of A09 and A12, and the tops of A05 and A08.

Figure 2 Chromosome location of GhPRX genes.

GhPRXs were located on 26 chromosomes of G. hirsutum, and four genes were found on scaffold.

Gene structural analysis of GhPRXs

To further explore the structural diversity of the GhPRX genes, we analyzed the exons and introns of the 166 GhPRX genes. The numbers of GhPRX gene exons varied from 1–8. Most members contained four exons (104/166, 62.6%, Fig. 3B). We identified a conserved intron/exon gene structure for the GhPRX genes. More than half of the GhPRX genes had three introns and four exons (103/166, 62.6%), highly similar to the Arabidopsis PRX gene structure (Tognolli et al., 2002). However, the gene structures of 52 GhPRX family members were inconsistent with the three intron/four exon structure. Their intron numbers changed during evolution. However, in the same subfamilies, most members shared great similarity in gene structures and numbers of exons.

Figure 3 Analysis of PRXs gene structure and cis-acting elements in G. hirsutum.

(A) Phylogenetic analysis of PRX genes (B) exon and intron structure analysis of PRX genes. Introns and exons are represented by thin lines and green boxes, respectively. (C) Cis-acting elements of PRX genes promoters. The UTR is shown in a yellow box.

Gene duplication analysis of GhPRXs

Gene duplication, including tandem duplication and segmental duplication, is the main driving force in the evolution of genomes (Cannon et al., 2004). In this study, we identified 121 paralogous gene pairs in G.hirsutum by BLASTP and MCScanX. Among them, 100 included segmental duplications, while the remaining 21 were tandem duplications (Fig. 4). Segmental duplication was likely to be the main reason for the expansion of the GhPRX gene family. In general, Ka/Ks < 1 indicates negative or purifying selection, Ka/Ks = 1 stands for neutral selection, and Ka/Ks > 1 suggests positive selection. The Ka/Ks ratios of the GhPRX gene pairs were <1 except for the GhPRX134 and GhPRX135 gene pair (Fig. 4, Table S5), implying that these gene pairs underwent purifying selection.

Figure 4 Gene duplication analysis of GhPRXs.

(A) Paralogous gene pairs among G. hirumtum. (B) Ka, Ks, Ka/Ks distribution of PRXs gene pairs. Ka, Ks, Ka/Ks analysis of GhPRXs-GhPRXs.

Analysis of GhPRX promoters

The upstream promoter regions of genes contain cis-acting elements that regulate gene transcription. Here we analyzed the sequences of 1,500 bp upstream of the GhPRX genes using Plantcare (Lescot et al., 2002). Based on their putative functions, the cis-acting elements were categorized into three major groups, i.e., the hormone-responsive, stress-responsive and growth-responsive cis-regulatory groups (Fig. 3C, Table S6). We found that the number of regulatory elements related to plant hormones was the largest, which included methyl jasmonate (MeJA; 300, 93/166, 56%), abscisic acid (ABA; 294, 122/166,73.49%), gibberellin (GA; 89, 67/166, 40.36%), salicylic acid (SA; 89, 66/166, 39.75%) and auxin (IAA; 33, 30/166, 18.07%). Among the plant hormonal cis-acting regulatory elements, the number of GhPRXs having at least one ABA element was the largest, and the total number of MeJA elements across GhPRXs was the largest. More than half of the GhPRXs contained one to sixteen MeJA cis-acting regulatory elements. The stress-response element was the second largest category, which includes drought (MBS)-, low temperature (LTR) - and defense-response elements. In addition, we also found cis-acting elements involved in endosperm, circadian, seed specific, anaerobic, meristem and cell cycle regulation. In summary, these results suggested that GhPRXs play important roles in plant growth, development and responses to abiotic stresses.

GO and KEGG enrichment analysis of GhPRXs

To further understand the functions of GhPRXs, we performed functional enrichment annotations of gene ontology (GO) using FDR ≤ 0.05 as the cutoff. A total of 265 GO terms were obtained, 51 out of which were significantly enriched (Table S7). The top 20 significantly enriched terms were visualized using the Omicshare tools (Fig. 5). The enriched gene ontology-biological processes (GO-BP) included the response to oxidative stress, and oxidation-reduction process and response. The gene ontology-molecular function (GO-MF) results showed the peroxidase activity, oxidoreductase activity and antioxidant activity enrichment. The gene ontology-cellular component (GO-CC) results suggested the GhPRX family genes were significantly enriched in the plant-type cell wall. Meanwhile, we carried out functional enrichment of Kyoto encyclopedia of genes and genomes (KEGG). We detected three signaling pathways in the KEGG analysis, among which the phenylpropanoid biosynthesis pathway was significantly enriched (Fig. 6, Table S7). Taken together, GhPRXs were involved in many biological processes, including the response to oxidative stress, peroxidase activity, and phenylpropanoid biosynthesis.

Figure 5 Bubble plot showing GO enrichment analysis of GhPRXs.

The top 20 GO terms significantly enriched by GhPRXs. Rich Factor indicates the ratio of the number of genes located in this GO term in GhPRX family genes to the total number of genes located in this GO term in all background genes. GeneNumber indicates the number of genes located in this GO term in GhPRX family genes.

Figure 6 Bubble plot showing KEGG enrichment analysis of GhPRXs.

Rich Factor indicates the ratio of the number of genes located in this GO term in GhPRX family genes to the total number of genes located in this GO term in all background genes. GeneNumber indicates the number of genes located in this GO term in GhPRX family genes.

Expression patterns of GhPRXs

We investigated the expression patterns of GhPRX genes using publicly available RNA-seq data of nine different tissues (bract, petal, torus, root, leaf, stem, pistil, sepal and anther). Due to the large number of members in the GhPRX family, genes were divided into two groups (A and B) based on the subfamily for analysis, and the expression was shown as log2 values (Fig. 7). The expression patterns of the A group (Fig. 7A) could be divided into five clades, namely clade 1 to 5. Clade 1 contained 29 genes with expression high in root. Clade 2 included 27 genes, including GhPRX143 and GhPRX152, which were not expressed in all the eight tissues. Clades 3 and 4 had 21 genes, most of which showed low expression levels in most tissues. The expression levels of clade 3 genes were lower than those of clade 4 genes. Clade 5 contained eight genes, which showed high expression levels in most tissues. Particularly, all the Clade 5 genes showed high expression levels in petals and anthers. The B group (Fig. 7B) showed similar expression patterns as the A group (Fig. 7A). GhPRX107 showed high expression levels in all the tissues, implying its essential role during plant development. GhPRX27 and GhPRX99 had similar tissue expression patterns, which indicated their similar functions.

Figure 7 Expression patterns of GhPRX genes in different tissues of G. hirsutum.

These genes were divided into two groups (A and B) based on subfamily for analysis, and the expression was shown as log2 values.

PRXs are important in plant fertility (Jacobowitz, Doyle & Weng, 2019). Therefore, we used transcriptomics data of our sterile line (JinA) and maintainer line (MB177) flower buds to explore the expression patterns of 166 GhPRX genes (Fig. 8). Using the same two groups as defined above, we detected the expression levels of seven genes were significantly different between the sterile and maintainer lines. Interestingly, the effect directions of the seven genes were different: GhPRX107, GhPRX27 and GhPRX99 were significantly up-regulated in the sterile line, while GhPRX44, GhPRX124, GhPRX48 and GhPRX128 were significantly down-regulated in the sterile line.

Figure 8 Expression patterns of GhPRX genes during flower bud development of sterility line-Jin A and mataintainer line-MB177.

These genes were divided into two groups (A and B) based on subfamily for analysis, and the expression was shown as log2 values (B2) microsporocyte stage of Jin A. (B3) Meiosis stage of Jin A. (K2) microsporocyte stage of MB177. (K3) meiosis stage of MB177.

To validate the transcriptomics data, we selected two genes (GhPRX107 and GhPRX128) from the seven differentially expressed genes and performed qRT-PCR in the sterile (JinA) and maintainer line (MB177) flower buds that were collected at different stages (sporogonium stage, microsporocyte stage and meiosis stage). Consistent with RNA-seq data, GhPRX107 was significantly up-regulated in the sterile line flower buds at all the three stages, while GhPRX128 showed the opposite effect (Fig. 9). Collectively these results indicated that these seven PRX genes are closely related to pollen fertility.

Figure 9 qRT-PCR results of GhPRX107 and GhPRX128 during flower bud development of sterility line (Jin A) and maintainer line (MB177).

(1) Sporogonium stage. (2) Microsporocyte stage. (3) Meiosis stage. Error bars showed the standard deviation of three biological replicates. Asterisks (**) show that the difference is extremely significant (P < 0.01).

Silencing GhPRX107 reduced ROS levels in microsporocyte-stage anthers

To explore the role of GhPRX107 in pollen fertility, we silenced GhPRX107 in cotton using virus-induced gene silencing (VIGS). After 11 days, the infected cotton leaves with TRV2:CLA1 showed photobleaching phenotype suggesting successful silencing of CLA1 (Fig. 10A). Using a similar strategy, we silenced GhPRX107 using TRV:GhPRX107. qRT-PCR results confirmed that GhPRX107 expression was significantly reduced in TRV:GhPRX107 plants compared to the control plants (TRV:00; Fig. 10B).

Figure 10 VIGS validates the function of GhPRX107.

(A) The phenotypes of TRV2:CLA1, TRV2:00 (empty load) and TRV2:GhPRX107 cotton seedlings. (B) The expression of GhPRX107 insilenced and control plants. Error bars showed the standard deviation of three biological replicates. Asterisks (**) show that the difference is extremely significant (P < 0.01). (C) Anthers stained with Nitroblue tetrazolium (NBT), show O2− level in TRV:00 and TRV:Ghprx107 plants. (D) Anthers stained with 3,3-diaminobenzidine, show H2O2 in TRV:00 and TRV:Ghprx107 plants. Bars = 200 um.

Previous studies showed that PRXs were not only oxidizing target substrates with H2O2, but also acting as key factors in producing ROS. To explore if GhPRX107 is associated with ROS production, we analyzed the ROS (O2− and H2O2) levels of anthers at the microsporocyte stage between the GhPRX107-silenced and control plants by staining with NBT and DAB, respectively. We showed that ROS (O2− and H2O2) levels were significantly decreased in the GhPRX107-silenced cotton plants compared to the control plants (Figs. 10C, 10D). Our results suggested an association between GhPRX107 expression levels and ROS levels in anthers.

GhPRX107 overexpression in Arabidopsis enhanced ROS levels in anthers

To further explore the role of GhPRX107 in male reproductive processes, we genetically transformed Arabidopsis using a GhPRX107 overexpression vector. NBT and DAB staining showed that the levels of superoxide anion (O2−) significantly increased around stages 6 and 7 in Arabidopsis anthers overexpressing GhPRX107. Hydrogen peroxide levels (H2O2) also significantly increased from stages 7 to 9 compared with the wild type (Fig. 11). These results further suggested an association between GhPRX107 expression levels and ROS levels in anthers during microspore development.

Figure 11 Comparison of ROS level between wild-type and overexpressing expression Arabidopsis anthers.

(A) NBT staining analysis of O2− in anthers at various developmental stages from the wild type and OE. (B) DAB staining analysis of H2O2 in anthers at various development from wild type and OE. Classification of anther (stage 6–12) is based on anther sizes. Bars = 200 um.

Discussion

Class III peroxidases are plant-specific. PRXs contain various kinds of isoenzymes and carry out different enzymatic reactions in life processes of plants. The PRX gene family plays an important role in biotic and abiotic stress response, and plant growth and development. Currently, the Class III peroxidase gene families of Arabidopsis thaliana (Tognolli et al., 2002), Populus, Oryza sativa (Passardi et al., 2004), Maize (Wang et al., 2015), Pear (Cao et al., 2016) and Brachypodium distachyon (Zhu et al., 2019) have been identified and analyzed.

In this study, we identified 166, 78 and 89 PRX genes in G. hirsutum, G. arboretum and G. raimondii, respectively. Compared with the PRX genes in Arabidopsis, we found more PRX genes in G. hirsutum. Although both Arabidopsis and G. hirsutum are dicotyledonous plants, the different degrees of polyploidy may be the main reason driving the different numbers of PRX genes in the two plants.

G. hirsutum is allotetraploid with A and D genomes. Previous studies have shown that the Dt subgenome of G. hirsutum came from G. raimondii. The A2 genome of G. arboretum and the At subgenome of G. hirsutum may originate from a common ancestor (Du et al., 2018). The collinearities were largely conserved between the At subgenome and the A2 genome, and between the Dt subgenome and the D5 genome. Specifically ~75.3% of the TM-1 At subgenome matched with 72.1% of the A2 genome in one-to-one syntenic blocks, and ~78.1% of the TM-1 Dt subgenome matched with 85.6% of the D5 genome in one-to-one syntenic blocks (Yang et al., 2019). PRX genes corroborated this relationship among the three cotton species. We constructed a maximum likelihood (ML) phylogenetic tree of PRXs in G. hirsutum, G. arboretum, G. raimondii and Arabidopsis and showed that the PRXs of G. hirsutum can be divided into seven subfamilies with genes from the three cotton species contributing to each subfamily. Previous studies have divided Arabidopsis into five subfamilies (Tognolli et al., 2002). The reason for the inconsistency was that homologous genes of AtPRX47, AtPRX64, AtPRX66, AtPRX21 and AtPRX12, and AtPRX47, AtPRX66 and AtPRX66 were divided into two clades each, resulting in two extra subfamilies.

We found that the number and physical properties of PRXs showed differences between diploid (G. arboretum and G. raimondii) and tetraploid cotton species (G. hirsutum) as well, suggesting independent evolution of their genomes. In a long-term evolutionary process, terminal repeats have made important contributions to the expansion of A genome scale, speciation and evolution (Yang et al., 2019). Abundant species-specific structural variations in gene regions have changed the expression of many important genes. Compared with G. raimondii, there were some unique structural variations in G. hirsutum, for example, there were the large fragment inversions in D09 chromosome and large inter-arm inversions in D12 chromosome. This indicated these variations occurred after polyploidization (Wang et al., 2019). Moreover, the species-specific gene families with relatively high proportion experienced more expansion or contraction in diploid D5 genome species (Yang et al., 2021). Therefore, different evolutionary pressures may be the reason for the differences between A2 genome and At subgenome, D5 genome and Dt subgenome.

Spatial and temporal expression patterns of PRX genes relate to their functions. Genome-wide gene expression analysis in Arabidopsis flowers showed that the members of PRX family were highly expressed in floral organs (Wellmer et al., 2004). Genes specifically or mainly expressed in plant floral organs were reported to be integral in floral organ development (Chen et al., 2009). PRX9 and PRX40 genes have been shown to be essential for normal Arabidopsis tapetum and microspore development (Jacobowitz, Doyle & Weng, 2019). Although cotton PRX Ghpod gene was found to be specifically expressed in flower buds and possibly involved in male development processes of angiosperms (Chen et al., 2009), no comprehensive expression patterns of PRX genes have been identified in different tissues and along male development processes.

Here we analyzed the expression patterns of GhPRX family genes using the transcriptomics data of nine tissues from a public database. We found many PRX members were highly expressed in anthers. More importantly, we investigated this gene family using our own transcriptomics data from a sterile line and a maintainer line. During flower bud development, we detected three GhPRX genes significantly up-regulated and four genes significantly down-regulated in the male sterile line. We validated our findings using qRT-PCR at three flower bud development stages. Therefore, we hypothesized that these genes played important roles during male reproductive processes of cotton.

Since ROS is closely related to male reproductive processes (Hu et al., 2011; Xie et al., 2014; Yang, Han & Huang, 2014a) and PRX functions, we carried out functional studies of one PRX gene-GhPRX107. We found that ROS contents of the microsporocyte-stage anthers from GhPRX107-silenced cotton plants were significantly decreased than that of controls. Overexpression of GhPRX107 in Arabidopsis significantly increased ROS levels in anthers. Taken together GhPRX107 is a determinant of ROS levels in anther.

Conclusions

In this study, we identified 166, 78 and 89 PRX genes from G. hirsutum, G. arboretum and G. raimondii respectively. We studied this family of genes using phylogenetic analysis, subcellular localization analysis, gene structure, gene duplication and cis-acting element analysis. We showed that most PRXs are conserved during the evolution process, and segmental duplication and purifying selection were the major drivers in the evolution of GhPRX gene family. Based on the transcriptome data analysis, we found that the expression levels of seven genes were significantly different between a sterile and a maintainer line, which suggested their involvement in pollen fertility. Importantly, silencing GhPRX107 decreased ROS contents of microsporocyte-stage anthers in cotton compared to controls. Overexpressing GhPRX107 enhanced ROS levels in anthers and changed the spatiotemporal pattern of ROS production in transgenic Arabidopsis plants. These results suggested an association between GhPRX107 expression levels and ROS levels in anthers. However, the relationship between GhPRX107 and male reproductive process of cotton needs further research. This study provides a useful reference for further analysis of the GhPRX gene family evolution and sets the foundation to study the potential functions of GhPRX genes in reproductive processes of cotton males.

Supplemental Information

Supplemental Information 1 Cotton genome GO and KEGG.

Click here for additional data file.

Supplemental Information 2 Primer used for qRT-PCR in this study.

Click here for additional data file.

Supplemental Information 3 Primer used for plasmid construction in this study.

Click here for additional data file.

Supplemental Information 4 List of the identified PRX genes in cotton.

Click here for additional data file.

Supplemental Information 5 Distribution of Ka, Ks, Ka/Ks of PRXs gene pairs.

Click here for additional data file.

Supplemental Information 6 The cis-acting elements of GhPRXs promoters.

Click here for additional data file.

Supplemental Information 7 The GO and KEGG positive hits.

Click here for additional data file.

Supplemental Information 8 Raw data: qRT-PCR results of GhPRXs during different flower bud development stages.

Click here for additional data file.

Additional Information and Declarations

Competing Interests

Author Contributions

Data Availability

The authors declare that they have no competing interests.

Yi Chen conceived and designed the experiments, performed the experiments, analyzed the data, prepared figures and/or tables, authored or reviewed drafts of the article, and approved the final draft.

Jiajia Feng performed the experiments, analyzed the data, prepared figures and/or tables, and approved the final draft.

Yunfang Qu conceived and designed the experiments, prepared figures and/or tables, authored or reviewed drafts of the article, and approved the final draft.

Jinlong Zhang performed the experiments, analyzed the data, prepared figures and/or tables, and approved the final draft.

Li Zhang performed the experiments, analyzed the data, prepared figures and/or tables, and approved the final draft.

Dong Liang performed the experiments, analyzed the data, prepared figures and/or tables, and approved the final draft.

Yujie Yang analyzed the data, prepared figures and/or tables, and approved the final draft.

Jinling Huang conceived and designed the experiments, authored or reviewed drafts of the article, and approved the final draft.

The following information was supplied regarding data availability:

The raw data is available in the Supplemental File.

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
