# Peer review of "Genome-wide identification and functional analysis of class III peroxidases in Gossypium hirsutum"

_PeerJ, doi:10.7717/peerj.13635_

## Round 0.1 · original submission · Major Revisions

Dear Dr. Chen:

Your manuscript has been reviewed by four experts in your research area and their comments/suggestions are enclosed. I would appreciate it if you could revise your manuscript and submit it at your most convenient time.

Below are some MAJOR comments:

1. "Discussion should be robust along with the findings and provide more insights instead of a repeated description of previous results".

2. Phylogenetic analysis is not clear. Please provide detailed information on the software you used.

3. The major shortcoming is that there is "no direct evidence in the data to support the authors' hypothesis that Class III peroxidase genes are linked to pollen fertility and pollen development". Please provide experimental evidence or revise your working hypothesis.

4. "The discussion is very descriptive". Please rewrite.

5. Comprehensive English editing is required before submitting your revised version of the manuscript. Please contact a professional editing service.

Best regards,

Tika Adhikari

Reviewer 1 ·

Basic reporting

Class III peroxidases are involved in various stages of plant growth and development. In this work, the authors investigated 166 PRX genes from G. hirsutum on genomic, transcriptomic and functional levels using combinatorial techniques, providing reference for further research on PRX function in plants.

Experimental design

For phylogenetic analysis, maximum likelihood method is usually recommended for publication purpose instead of neighbor-joining.

Validity of the findings

1. Line 199, could the authors explain how did they come to the conclusion that "the homologous gene pairs of GhPRXs in subgenome A and D mainly exhibited parallel evolution" from Fig 1?

2. Line 205-219, descriptions on group size and number of protein in each group are less relevant, which don't provide much insights. Clustering of some Arabidopsis and cotton PRX genes in the phylogenetic tree could just be resulted from some highly conserved PRXs across plants, doesn't necessary indicate any overall close relation of PRX in this two species from evolution perspective.

3. Fig. 4, by definition, two genes are said to be paralogous if they are derived from a duplication event, but orthologous if they are derived from a speciation event. Apparently the description here is inaccurate.

4. Line 282, for genes that seem to be not expressed, besides loss of function, more often, the reason could be that the expression levels are too low to be detected. Also, gene expression analysis is high restricted to the quality of RNA-seq including sample preparation.

5. Line 301 and Fig. 10, both GhPRX107 and GhPEX128 are higher expressed in sterility line than maintainer line, why does the authors mean by saying "opposite"?

6. ROS level can be affected by many factors, the speculation PRX107 is involved in reproductive process by regulating ROS status lacks solid evidence.

7. Discussion part should dig deeper into the findings and provide more insights instead of repeated description of previous results.

8. Typos and grammars need to be corrected (line 45, 101,104, 176, 190, 195, 201, 222, 228, 274, 279, 289, 302, 317, et al).

Reviewer 2 ·

Basic reporting

The submitted manuscript analyzes RPX genes, which have been studied in various plants including maize, pears, and Populus. Here the authors specifically focused on one of the unstudied plants, Gossypium hirsutum. The analyses presented here largely follow Zhu et al (2019, https://pubmed.ncbi.nlm.nih.gov/30904716/), with additional analyses on the associations between RPX genes and male fertility. I think in general this is a nice effort. Some suggestions for improvement are provided below.

There are quite a lot of awkward sentences (e.g. lines 33, 38, 190, 199, 244), typos, and grammatical errors (e.g. lines 197, 227, 245, 301, 305) throughout the manuscript, which poses difficulty for readers to clearly understand the text. I suggest the authors use copy editing services or let someone proficient in English to proofread your manuscript. In addition, Fig 10 and Fig 11B need to have higher quality.

Experimental design

In general, the methods for bioinformatic analyses need to be described in much more details so that the readers can reproduce your analyses. In particular, you need to provide the parameters you used for all the bioinformatic analyses you have done. You also need to cite the tools you use correctly: for example, for line 129-132, you should cite the article “PlantCARE, a database of plant cis-acting regulatory elements and a portal to tools for in silico analysis of promoter sequences” in addition to the website. For GO and KEGG analyses, you will also need to cite the original papers for the databases.

Here are some specific sections that require more details in particular:
Phylogenetic analysis: what specific tools did you use to align the PRXs protein sequences? What parameters did you use? In addition, how did you cluster the genes into different subgroups?

Gene structure analysis: you need to explain what GFF3 profiles are. Also, how did you get the GFF3 files? For the tool you used, what parameters did you choose?

Gene duplication and calculation of Ka/Ks values: how did you find the collinearity gene pairs? Or did you find them using MCScanx? The one sentence explanation in line 124 does not clearly describe what you did.

Analysis of GO and KEGG pathway enrichment: for both GO and KEGG analyses, what background genes and how many did you use? Did you use multiple testing corrections for your p-values?

Expression pattern analysis: write the full name of FPKM. More importantly, how did you process the RNA-seq data to calculate FPKM? What tools and parameters did you use?

Validity of the findings

I suggest the authors to revisit GO and KEGG analyses to make sure they are correct. Here are some specific comments that may help you improve the quality and soundness of your manuscript:

1. Line 183-185: are there any duplicate PRX genes within this set of 166 genes? Are they non-redundant? It would be nice to check the number of non-redundant genes.
2. Line 185-187: I believe these are predicted numbers as you don’t actually have the encoded proteins. As in Fig 8, some GhPRX genes are not expressed. It would be better to explicitly say it clearly to avoid confusion. Same with sentences in line 190-194.
3. Line 188: can you say something quantitative about the similarity? It is difficult to say GhPRXs share high similarity to the maize PRXs proteins without numbers. In addition, how did you compare the similarities between two sets of proteins? I don’t find them in Methods.
4. Fig 1: “… four genes are located on scaffold” is not very accurate because all GhPRXs are located on 26 chromosomes. “Found” would be a better word as the scaffolds are results of fragmented assembly.
5. Line 199-200: can you explain more about parallel evolution? The sentence is difficult to understand probably due to its awkwardness.
6. Line 208-209: how did you determine eight subgroups? Can you add the explanation in Methods?
7. Fig 2: what do the colors for check marks and stars represent?
8. Fig 4: (A) It’s only paralogous gene pairs right, as it is within G. hirsutum? (B) The figures do not make sense to me. The x- and y-labels are exactly the same. What are they exactly?
9. Line 239: < 1 Ka/Ks indicates possible purifying selection, but it does not suggest strong purifying selection. Removal of word “strong” will be better.
10. Fig 5: is (A) is the same as Fig 3(A)? If so, you can probably combine Fig 3 and Fig 5 into one figure.
11. Line 246: can you list the number/percentage of regulatory elements related to plant hormones? It is too hard to tell from Fig 5. In addition, you probably need to a supplementary table for this analysis so readers can easily find regulatory elements of GhPRX genes.
12. Line 258-261: full name of analyses should be listed together with the abbreviations because not all readers are familiar with such kind of analyses. For example, you should say biological process explicitly for GO-BP. Same for GO-MF and GO-CC.
13. Line 257-264: how many GO terms out of the total are statistically significant? Fig 6 only shows the top 20. You should report all positive hits in a supplementary table.
14. Fig 6 and Fig 7: explain what rich factor and GeneNumber mean. The legend for GeneNumber in Fig 7 does not make sense – two circles with different sizes have the same numeric value of 33.
15. Line 256-270: I would re-examine the GO and KEGG analyses again to make sure they are correct. The results don’t look right to me. In particular, both Fig 6 and Fig 7 have extremely low p/q-values. For example, in Fig 6, the lowest reported -log10(p-value) = 200, which means p-value = 10^-200. p-value of 10^-2 = 0.01 is already quite low, and your p-values are extremely small (I don’t think you can calculate p-value to that precision). In Fig 7, is there a reason why you use q-value instead of p-value?
16. Fig 8: you may want to mention that you split all GhPRXs into two heatmaps as it is confusing to see two figures at the first glance. Besides, please label the legend. What is the unit for expression? Same for Fig 9.
17. Line 275-284: instead of grouping the expression profiles by hand, you may cluster the expression profiles and see how many groups there are.
18. Line 298-301: is it statistically significant? Can you use statistical test to make your statement more convincible? You may also want to label the significance in Fig 10.
19. Line 309: similar to comment 18, statistical test is needed for both text and Fig 11(B).
20. Fig 10 and Fig 11(B): what do A and B on top of each bar mean?

Reviewer 3 ·

Basic reporting

no comment

Experimental design

no comment

Validity of the findings

The major for the article is that there is no directly evidence in the data to support the authors' hypothesis that Class III peroxidase genes are linked to pollen fertility and pollen development. I would suggest the authors to retract that claim or provide further experimental evidence for that claim.

Additional comments

no comment

Reviewer 4 ·

Basic reporting

no comment

Experimental design

no comment

Validity of the findings

no comment

Additional comments

This study is very interesting and combines bioinformatics and experimental analysis to provide information on the GhPRX gene family in Gossypium hirsutum. Generally, the manuscript is well structured but needs some improvements before publication. I pointed out several comments and suggestions to help you to improve your manuscript:

M&M
- Did you use MUSCLE or CLUSTALW to perform the alignments? Which parameters did you use? These informations should be provided.
- What substitution model did you use in the NJ analysis? How did you treat the gaps and missing data?
- I would suggest you perform the phylogenetic analysis with Bayesian or ML methods.
- Provide more details of this gene structure analysis.
- Did you test if the EF-1α gene is efficient to be used as a reference gene in this experiment? I would recommend testing more genes or providing the reference that has previously tested it.

RESULTS
- How was the length of the alignment used for the phylogenetic reconstruction? Did you consider codifying the extended gaps before the analysis?
- I suggest including the genes identified in G. raimondii in the phylogeny to compare the groups within the genus Gossypium?

The discussion is very descriptive. You should make an effort to better discuss the results.

---

## Round 0.2 · Minor Revisions

Dear Dr. Chen:

Thank you for submitting the revised manuscript. Your manuscript has been further reviewed by the two reviewers and there are still few comments (see revewer # 2 comments). Please revise and submit it at your the earliest possible time. Thank you.

Best regards,

Sincerely,

Tika Adhikari

Reviewer 1 ·

Basic reporting

The authors have well addressed comments from reviewers.

Experimental design

Well.

Validity of the findings

Well.

Additional comments

No

Reviewer 2 ·

Basic reporting

The English is still awkward -- the manuscript has a lot of awkward/grammatically incorrect sentences that need improvement (e.g. lines 130, 136, 250). Some references are missing for softwares used. For example, I think the authors use CLUSTALW (which they incorrectly spelled as CLUATALW) but they did not cite it.

Experimental design

no comment

Validity of the findings

The methods are more detailed than last time but I still don't know what kind of statistical tests they used.

Additional comments

Figure 3: legend is illegible
Figure 4B: histogram is a better choice for visualizing distributions of a single variable.
Figure 9, 10B: what statistical tests do you use and what significance level is for **?

---

## Round 0.3 · accepted · Accept

Dear Dr. Chen,
Thank you for your submission to PeerJ.
I am writing to inform you that your manuscript - Genome-wide identification and functional analysis of class III peroxidases in Gossypium hirsutum - has been Accepted for publication. Congratulations!

Best regards,

Tika Adhikari

Another Section Editor, Pedro Silva, noted:

> in the legend to fig. 9, I believe "** show that the difference is extremely significant." should be changed to "** show that the difference is extremely significant (P < 0.01)."